# Combination of *Lactobacillus plantarum* HAC03 and *Garcinia cambogia* Has a Significant Anti-Obesity Effect in Diet-Induced Obesity Mice

**DOI:** 10.3390/nu15081859

**Published:** 2023-04-12

**Authors:** Youn-Goo Kang, Taeyoung Lee, Jaeyoung Ro, Sanghun Oh, Jin-Hwan Kwak, Ah-Ram Kim

**Affiliations:** 1School of Creative Convergence Education, Handong Global University, Pohang 37554, Gyeong-Buk, Republic of Korea; ygkang@handong.ac.kr; 2School of Life Science, Handong Global University, Pohang 37554, Gyeong-Buk, Republic of Korea; 3HDSbio Inc., Pohang 37668, Gyeong-Buk, Republic of Korea; 4Sunlin University, Pohang 37560, Gyeong-Buk, Republic of Korea; 5School of Applied Artificial Intelligence, Handong Global University, Pohang 37554, Gyeong-Buk, Republic of Korea

**Keywords:** obesity, probiotics, lactobacillus plantarum, garcinia cambogia, synergic effect, gut microbiota

## Abstract

Obesity is a major global health problem which is associated with various diseases and psychological conditions. Increasing understanding of the relationship between obesity and gut microbiota has led to a worldwide effort to use microbiota as a treatment for obesity. However, several clinical trials have shown that obesity treatment with single strains of probiotics did not achieve as significant results as in animal studies. To overcome this limitation, we attempted to find a new combination that goes beyond the effects of probiotics alone by combining probiotics and a natural substance that has a stronger anti-obesity effect. In this study, we used a diet-induced obesity mouse (DIO) model to investigate the effects of combining *Lactobacillus plantarum* HAC03 with *Garcinia cambogia* extract, as compared to the effects of each substance alone. Combining *L. plantarum* HAC03 and *G. cambogia,* treatment showed a more than two-fold reduction in weight gain compared to each substance administered alone. Even though the total amount administered was kept the same as for other single experiments, the combination treatment significantly reduced biochemical markers of obesity and adipocyte size, in comparison to the treatment with either substance alone. The treatment with a combination of two substances also significantly decreased the gene expression of fatty acid synthesis (FAS, ACC, PPARγ and SREBP1c) in mesenteric adipose tissue (MAT). Furthermore, 16S rRNA gene sequencing of the fecal microbiota suggested that the combination of *L. plantarum* HAC03 and *G. cambogia* extract treatment changed the diversity of gut microbiota and altered specific bacterial taxa at the genus level (the *Eubacterium coprostanoligenes* group and *Lachnospiraceae* UCG group) and specific functions (NAD salvage pathway I and starch degradation V). Our results support that the idea that the combination of *L. plantarum* HAC03 and *G. cambogia* extract has a synergistic anti-obesity effect by restoring the composition of the gut microbiota. This combination also increases the abundance of bacteria responsible for energy metabolism, as well as the production of SCFAs and BCAAs. Furthermore, no significant adverse effects were observed during the experiment.

## 1. Introduction

A major health problem, obesity is defined by the World Health Organization (WHO) as a condition in which an abnormal or excessive accumulation of fat negatively impacts health [1]. According to the WHO, obesity is one of the top 10 risk factors for health problems [2]. More than 1.9 billion people (more than 25% of the world’s population) were overweight in 2016, with more than 650 million being obese (8.71% of the world’s population) in 2016 [3]. Obesity is associated with many diseases, such as type 2 diabetes, dyslipidemia, hypertension, fatty liver, coronary artery disease, colorectal cancer, as well as psychological conditions, such as psychological atrophy or depression [4].

Three chemical-based obesity treatments were approved by both the US Food and Drug Administration (FDA) and the European Medication Agency (EMA) in 2021: liraglutide, orlistat, and a combination of liraglutide and bupropion [5]. Among these, liraglutide received FDA approval in 2014, and EMA received approval in 2015 [6]. Since then, liraglutide has maintained the top position in the global obesity treatment market. Liraglutide (product name Saxenda) is an analog of glucagon-like peptide-1 (GLP-1), which binds to and signals the GLP-1 receptor. GLP-1 is a hormone naturally produced by the body in response to food intake [7]. By increasing insulin secretion, the pancreas removes excess blood sugar, and glucagon secretion is reduced, thus controlling sugar levels in the blood stream. It plays an important role in maintaining normal blood sugar, and it binds to a receptor in the brain and transmits a signal to the hypothalamus of the brain, thereby reducing and suppressing appetite [8]. However, it was reported that there were side effects in other tissues such as the liver [9], pancreas [10] and heart [11], as well as adverse effects in the kidneys such as nausea, diarrhea and vomiting [12]. Moreover, it has been reported that the incidence of some cancers such as breast cancer [13] and pancreas cancer [14] increased in specific cases.

Among the various attempts to overcome the limitations of current chemical treatments, microorganism-based treatments are attracting worldwide attention [15]. Human organisms and microorganisms are inseparable. A typical 70 kg adult male consists of 30 × 10^13^ human cells and 38 × 10^13^ microorganisms [16]. The largest number of bacteria in the human body are found in the digestive tract, which begins at the mouth and extends to the stomach, small intestine, large intestine, and rectum, where 4 × 10^13^ bacteria are present. A number of studies have indicated that dysbiosis can induce obesity [17,18,19]. The gut microbiota of obese patients was generally less diverse when compared to that of healthy people [20]. *Firmicutes* increased and *Bacteroidetes* decreased in phylum level, *Erysipelotrichaceae* increased, while *Bacteroidaceae*, *Lachnospiraceae*, and *Eubacteriaceae* decreased at a family level [21,22,23]. As a result of these findings, microbiomes are currently being researched around the world as candidates for curing dysbiosis and possibly alleviating or curing diseases in the future [24].

Lactic acid bacteria (LAB) constitute a diverse group of gram-positive, non-spore forming, facultatively aerobic, catalase-negative bacteria producing lactic acid as the main end-product of carbohydrate fermentation [25]. The LABs include various types of bacterial genera, such as *Streptococcus*, *Lactococcus* and *Lactobacillus* [26]. *Lactobacillus* spp. is one of the most actively studied and used probiotics since *Lactobacillus acidophilus* was first identified in infant feces by Ernst Moro in 1900 [27]. A number of studies have reported that lactic acid bacteria have a potential for anti-obesity and anti-metabolic disease [28,29,30]. *Lactobacillus plantarum* HAC03, which was used in this study, is a strain identified from the white kimchi, which is one of the traditional fermented foods of South Korea. Its potential as a probiotic has been verified through safety and functional verification [31]. While microorganism-based obesity treatments, including probiotics, are much safer than chemically based treatments, they have not yet demonstrated enough efficacy to replace chemical-based treatments. To overcome this limitation, our research team researched the natural product that creates a synergistic effect with probiotics to provide a superior anti-obesity effect compared to probiotics alone.

Globally, plant-based weight control has been used since ancient times [32]. In numerous studies worldwide, *Garcinia cambogia* has been shown to have anti-obesity effects and is now being used as a weight loss product [33]. Hydroxycitric acid (HCA) has been identified as a key component of garcinia’s weight loss properties [34]. There are two major ways in which HCA affects weight loss. First, HCA has a structure similar to citric acid. In the pathway for making fat from sugar, ATP-citrate lyase (ACL) combines with citric acid to form acetyl CoA. HCA is used instead of citric acid to inhibit the subsequent synthesis of cholesterol or lipid [35]. Second, HCA suppresses appetite by increasing the level of serotonin in the brain and affects weight loss [36]. Consuming *G. cambogia* extract containing HCA has been observed to promote weight loss as well as alter gut microbiota composition, potentially inducing an anti-obesity effect via modulating the gut microbiota [37]. However, further research is needed to verify this claim. We hypothesize that a combination of a probiotic strain which has anti-obesity effects and *G. cambogia* with proven anti-obesity will exhibit a synergistic effect between the two substances, resulting in a greater anti-obesity effect than when each substance is used alone.

In this study, firstly the anti-obesity effect of *L. plantarum* HAC03 was investigated, and secondly, the anti-obesity effect of a combination of *L. plantarum* HAC03 and *G. cambogia* extracts, a natural compound that has already demonstrated anti-obesity properties, was investigated.

## 2. Materials and Methods

### 2.1. Bacterial Strains, Culture Conditions and Natural Product

*L. plantarum* HAC03 was grown in MRS broth (Difco Laboratories INC., Franklin Lakes, NJ, USA) at 37 °C. For the administration to mice, bacterial cells were dissolved in 1x PBS with two concentrations. To verify the effect of the bacteria itself, the bacteria were suspended at a concentration of 1 × 10^9^ CFU/mouse, and in the experiment to confirm the synergistic effect of the bacteria and the *G. cambogia* extract, the bacteria were suspended at a concentration of 1 × 10^8^ CFU/mouse. *Lactobacillus plantarum* 299V was used as a positive control to compare the effect of *L. plantarum* HAC03. *G. cambogia* extracts (main component: hydroxycitric acid, 65%) was obtained from DAEUNE FS Co., Ltd. (Ansan-si, Gyeonggi-do, Republic of Korea).

### 2.2. Animal Experiments

Animal experiments were performed in accordance with protocols approved by the Committee on the Ethics of Animal Experiments of the Handong Global University (Permit number: 20221026-12). Five-week-old C57BL/6 male mice from Saeronbio Inc. (Uiwang-si, Gyeonggi-do, Republic of Korea) were housed at 23 ± 1 °C and 60 ± 5% humidity, on a 12 h light/dark cycle. After 1 week of adaptation, the mice were divided into 8 groups. (*N* = 7 per group). Table 1 provides a detailed description of the groups. Before the oral administration of bacteria or *G. cambogia* extract, LFD (10% kcal from fat, D12450J, Research Diets Inc., New Brunswick, NJ, USA) or HFD (60% kcal from fat, D12492, Research Diets Inc., New Brunswick, NJ, USA) was given for a week to induce dysbiosis. The diet was changed to a high-fat diet or a low-fat diet 1 week before the probiotics treatment was conducted, since starting to take a high-fat diet and probiotics treatment at the same time may not have a significant effect [38]. Animal experiments were conducted for two purposes. In the first experiment, to verify the anti-obesity effect of *L. plantarum* HAC03, 200 µL PBS or 1 × 10^9^ CFU/mouse of probiotics (*L. plantarum* 299V or *L. plantarum* HAC03) were orally administered once a day for 10 weeks, depending on the group. In the second experiment, in order to verify the synergistic effect of *L. plantarum* HAC03 and *G. cambogia* extract, 1 × 10^8^ CFU/mouse of *L. plantarum* HAC03 and 200 mg/kg of *G. cambogia* extract containing 65% HCA was orally administered alone or in combination for 11 weeks. On the last day of the experiment, the mice were sacrificed by CO_2_ gas. The organs (liver, spleen, small intestine, colon, subcutaneous adipose tissue (SAT), epididymal adipose tissue (EAT), mesenteric adipose tissue (MAT) and brown adipose tissue (BAT)) were collected and stored at −80 °C until used.

### 2.3. Serum Analysis

Blood was collected through cardiac puncture and collected in a blood collection tube with lithium heparin (Becton, Dickinson and Company, Franklin Lakes, NJ, USA). It was centrifuged for 15 min at 2000× *g*, 4 °C to isolate the serum. Biochemical parameters in serum, including total cholesterol and triglycerides, were determined using a 7180 Clinical Analyzer (Hitachi High-Tech, Tokyo, Japan).

### 2.4. Histological Analysis

Part of the adipose tissue was fixed in 10% *v*/*v* formalin/PBS for histology analysis. It was embedded in paraffin for staining with hematoxylin and eosin. Images were obtained using a ZEISS Axio Imager 2 (Carl Zeiss Co., Ltd., Land Baden-Württemberg, Germany) at a magnification of 200×. The areas of adipocytes were measured using the ImageJ software with an Adiposoft plug-in, according to the developer’s instruction.

### 2.5. Real-Time RT PCR

The total RNA was extracted using TRIzol Reagent (Invitrogen., Waltham, MA, USA). cDNA was synthesized using a SuperiorScript III cDNA Synthesis kit (Enzynomics., Inc, Daejeon, Republic of Korea). Quantitative PCR was measured with an Applied Biosystem StepOnePlusTM Real-Time PCR system (Applied Biosystems, Waltham, MA, USA) using TOPrealTM qPCR 2X PreMIX (SYBR Green with high ROX) kit (Enzynomics., Inc., Daejeon, Republic of Korea). All the primers used for qPCR were synthesized by Macrogen (Seoul, Republic of Korea). For the qPCR primer sequence used in this study, refer to Table 2. Results were presented as mean ± S.D. normalized to expression of β-actin using the ΔΔCt method, in which the HFD group was used as the reference group.

### 2.6. Gut Microbiota Analysis

DNA was extracted from fecal samples using the AccuStool DNA Preparation kit (AccuGene, Incheon, Republic of Korea) in accordance with the manufacturer’s instructions. The hypervariable V3-V4 region of the 16S-rRNA gene was amplified from the DNA extracts through 25 PCR cycles using KAPA HiFi HotStart ReadyMix (Roche, Basel, Switzerland) and barcoded fusion primers 341f/805r containing Nextera adaptors [39]. PCR products (~428 bp) were purified with HiAccuBeads (AccuGene, Incheon, Republic of Korea). The amplicon libraries were pooled at an equimolar ratio and the pooled libraries were sequenced on an Illumina MiSeq system using MiSeq Reagent Kit v3 for 600 cycles (Illumina, San Diego, CA, USA). All raw data sets were denoised by correcting amplicon errors and were used to infer exact amplicon sequence variants (ASVs) using DADA2 v1.16 [40]. The SILVA release 138 rRNA reference database was used to create a Naïve Bayes classifier in order to classify the ASVs obtained from DADA2 [41]. Downstream analyses of quality-filtered and chimera-filtered reads were performed using the QIIME2-2022.8 software package [42]. Each of the sequences obtained from the DADA2 datasets was assigned to a taxonomy with a threshold of 99% pairwise identity using QIIME2 workflow scripts and the SILVA release 138 rRNA reference database classifier. Gut microbiota analysis, including alpha and beta diversity and relative abundance of gut microbiota, was performed using Microbiome [43]. The functional potential of each group based on the bacterial taxonomy was predicted using the PICRUSt2 [44]. Using LEfSe (Linear discriminant analysis effect size), the relative quantity of bacteria in each group was compared to find group-specific strains [45].

### 2.7. Statistical Analyses

Except for microbiota, statistical analyses were performed using Origin2022b (OriginLab., Northampton, MA, USA). The experimental results were presented as means ± standard deviation (S.D) for 7–8 mice in each group and analyzed with one-way ANOVA with Dunnett’s test compared with different groups. A *p* value of <0.05 was considered to be statistically significant.

## 3. Results

### 3.1. L. plantarum HAC03 Has an Anti-Obesity Effect on Dietary-Induced Obesity Mice

To confirm the anti-obesity effect of *L. plantarum* HAC03 alone, *L. plantarum* HAC03 was administered orally for 10 weeks at a concentration of 1 × 10^9^ CFU/mouse in the DIO model (Figure 1a). During the 10-week animal experiment, no significant adverse effects were observed. After 10 weeks of the animal experiment, the high-fat diet group (HFD) gained significantly more weight than the low-fat diet group (LFD) (Figure 1b). As compared to the HFD group, the HAC03 treatment group (HFD + L1) showed a similar reduction in weight gain as the 299V treatment group (HFD + L2) used as a positive control. There was a significant reduction in adipose tissue weight in the HFD + L1 group, including subcutaneous adipose tissue (SAT), mesenteric adipose tissue (MAT) and brown adipose tissue (BAT), except for epididymal adipose tissue (EAT) (Figure 1c,d). The results show that *L. plantarum* HAC03 has anti-obesity properties.

### 3.2. Combining L. plantarum HAC03 with G. cambogia Extract Has a Synergic Effect on Losing Weight

In order to determine whether combined administration of *L. plantarum* HAC03 and *G. cambogia* extract had a greater effect than the sum of their individual effects, *L. plantarum* HAC03 concentration was diluted to 1/10 (1 × 10^8^ CFU/mouse) and *G. cambogia* extract was administered at a dose of 200 mg/kg lower than the dose used for weight loss (from 307.5 to 941.7 mg/kg) [46,47]. After treatment of *L. plantarum* HAC03 and *G. cambogia* extract alone or in combination for 11 weeks, 1 × 10^8^ CFU/mouse *L. plantarum* HAC03 treatment group (HFD + L1′) and 200 mg/kg *G. cambogia* extract treatment group (HFD + G) had no significant weight loss compared with the HFD group (Figure 2b,c). However, comparing the combined same concentration of *L. plantarum* HAC03 and *G. cambogia* extract group (HFD + L1′ + G), a significant decrease in weight gain was observed. As a result of measuring the weight of each adipose tissues, no significant differences were observed in the HFD + L1′ group and the HFD + G group compared to the HFD group. In contrast, the HFD + L1′ + G group showed significant weight loss in SAT, MAT and BAT except for EAT. In addition, the weight of adipose tissues in HFD + L1′ + G group decreased significantly when compared to the single administration groups (Figure 2d). During the treatment period, the HFD + L1′ + G group consumed significantly fewer calories per week than the HFD group, as well as the HFD + L1′ group and the HFD + G group (Figure 2e). According to the results, the combined effects of *L. plantarum* HAC03 and *G. cambogia* extract were greater than the sum of their individual effects.

### 3.3. Combining L. plantarum HAC03 and G. cambogia Has a Synergic Effect on Decreasing the Biochemical Markers of Obesity

In order to verify the anti-obesity effect according to the treatments more clearly, biochemical markers (alanine aminotransferase (ALT), aspartate aminotransferase (AST), total cholesterol (T-chol), triglyceride (TG), low density lipoprotein cholesterol (LDL), high density lipoprotein cholesterol (HDL)) related to obesity were analyzed in the serum. According to the serum analysis, all indicators except HDL decreased significantly compared to the HFD group (Table 3). In particular, the HFD + L1′ + G group significantly reduced ALT and AST levels compared to single administrations (HFD + L1′ and HFD + G). The HFD + L1′ + G group showed a significant reduction in T-chol and LDL levels compared to the HFD + G group. These results confirmed that the combination of *L. plantarum* HAC03 and *G. cambogia* extract had a greater effect than the sum of their individual effects.

### 3.4. Combining L. plantarum HAC03 and G. cambogia Has a Synergic Effect on Decreasing the Adipocyte Size

In order to confirm that changes had occurred in adipocytes, four adipocytes (subcutaneous adipose tissue (SAT), epididymal adipose tissue (EAT), mesenteric adipose tissue (MAT), and brown adipose tissue (BAT)) were stained with H&E and viewed under a microscope. All adipocytes in the HFD group were larger than those in the LFD group. In SAT, a significant decrease in adipocyte size was observed in the HFD + G group and in the HFD + L1′ + G group than the HFD group. In particular, the HFD + L1′ + G group significantly reduced adipocyte size as compared to single administrations (HFD + L1′ and HFD + G) (Figure 3a,b). In MAT, a significant decrease in adipocyte size was observed in the HFD + G group and the HFD + L1′+ G group than the HFD group. In particular, the HFD + L1′ + G group significantly reduced adipocyte size as compared to the HFD + L1′ group and the HFD + G group (Figure 3e,f). In BAT, a significant decrease in adipocyte size was observed in the treatment groups (HFD + L1′ group, HFD + G group and HFD + L1′ + G group). Contrary to SAT and MAT, the HFD + L1′ group reduced the size of adipocyte more than the HFD + L1′ + G group (Figure 3g,h). By contrast, in EAT, there were no significant differences in the size of adipocytes in the HFD + L1′, HFD + G, and HFD + L1′ + G groups compared with the HFD group (Figure 3c,d). Adipocyte size and adipose tissue weight showed similar trends by group. According to these results, even at the cellular level, the combination of *L. plantarum* HAC03 and *G. cambogia* extract had more anti-obesity effects than single administration.

### 3.5. Combining L. plantarum HAC03 and G. cambogia Has a Synergic Effect on Decreasing mRNA Expression of Fatty Acid Synthesis

Expression levels of mRNA involving fatty acid synthesis in the mesenteric adipose tissue (MAT) such as fatty acid synthase (FAS), acetyl-CoA carboxylase (ACC), peroxisome proliferator activated receptor gamma (PPARγ), sterol regulatory element binding protein 1c (SREBP1c) were down-regulated in the LFD group compared with the HFD group (Figure 4). In all groups administered with *L. plantarum* HAC03 and G. cambogia extract alone or together (HFD + L1′, HFD + G and HFD + L1′ + G), the expression levels of all genes related fatty acid synthesis decreased compared to the HFD group. In the group that administered *L. plantarum* HAC03 and G. cambogia extract simultaneously (HFD + L1′ + G), the expression levels of ACC, PPARγ, and SREBP1c decreased compared to the group that received *L. plantarum* HAC03 (HFD + L1′) and *G. cambogia* extract (HFD + G) separately. These results confirmed that the combination of *L. plantarum* HAC03 and *G. cambogia* extract had a greater effect than the sum of their individual effects in mRNA expression of fatty acid synthesis in MAT.

### 3.6. Combining L. plantarum HAC03 and G. cambogia Affect Gut Microbiota Composition

Through numerous studies conducted worldwide, it has been revealed that there is a strong correlation between changes in the composition of gut microbiota and obesity [48,49]. To investigate the correlation between the anti-obesity effect of *L. plantarum* HAC03 and *G. cambogia* extract complex and the gut microbiome, we analyzed fecal samples to examine the composition of gut microbiota. According to the analysis of alpha diversity, the observed OTUs in the HFD group decreased compared to the LFD group, and the Shannon index decreased. The HFD + L1′ group showed an increase in both OTUs and Shannon index compared to the HFD group. However, the HFD + G group showed a significant decrease in alpha diversity, and a large variation was observed among samples (Figure 5a). Using the principal coordinate analysis (PCoA) based Bray–Curtis distance matrix method, beta diversity was analyzed, revealing differences between the HFD and LFD groups. PCoA1, PCoA2, and PCoA3 explained 32%, 18.79%, and 12.69% of the variance of the abundance of gut microbiota, respectively (Figure 5b). Figures using PCoA1 and PCoA2 or PCoA1 and PCoA3 as axes showed that the group that administered *L. plantarum* HAC03 or *G. cambogia* extract (HFD + L1′, HFD + G, and HFD + L1′ + G) was completely separated from the LFD or HFD groups, and these three groups were kept close together. However, in Figure 5b using PCoA2 and PCoA3 as axes, the three treatment groups were positioned between the LFD and HFD groups, and the group that combined *L. plantarum* HAC03 and *G. cambogia* treatment was closer to the LFD group. These results suggest that the combination of *L. plantarum* HAC03 and *G. cambogia* extract treatment can better restore dysbiosis associated obesity than administering them separately. 

To further observe the effects of the combination of *L. plantarum* HAC03 and *G. cambogia* extract on changes in gut microbiota, linear discriminant analysis effect size (LEfSe) was used to identify specific gut microbiota at the genus level according to the group (Figure 5c). In the HFD + L1′ + G group, the *Eubacterium coprostanoligenes* group and *Lachnospiraceae* UCG 010 group were significantly increased compared to other groups (Figure 5d). In addition, at the genus level, the HFD + L1′ group showed a significant increase in *Butyricicoccus*, while the HFD + G group showed a significant increase in *Bacteroides*, *Parabacteroides*, and *Defluviitaleaceae* UCG 011 (Appendix A). In particular, *Defluviitaleaceae* UCG 011 showed a significant increase in the LFD and HFD + G groups compared to the HFD group. Our results show that the distribution of gut microbiota can be different depending on whether *L. plantarum* HAC03 and *G. cambogia* extract are administered individually or in combination, and we have identified bacteria that significantly differ depending on the group.

To observe the effects of changes in gut microbiota distribution in combination of *L. plantarum* HAC03 and *G. cambogia* extract treatment on function, the functional abundances of each group based on 16S rRNA gene was predicted. HFD + L1′ + G group significant increase in the NAD salvage pathway I and starch degradation V functions (Figure 5e,f). In addition, based on the prediction of functional difference between five groups, HFD + L1′ group increased in branched chain amino acid synthesis functions, such as L-isoleucine biosynthesis, and pyruvate fermentation to isobutanol (Appendix A), while HFD + G group showed a significant increase in super-pathway of thiamin diphosphate biosynthesis II, thiazole biosynthesis I, and D-galacturonate degradation functions (Appendix A). According to these results, it is inferred that the combination of *L. plantarum* HAC03 and *G. cambogia* extract enhances specific functions through changing gut microbiota composition, which helps to restore the imbalance of gut microbiota associated with a high-fat diet, and thus plays a role in alleviating obesity.

## 4. Discussion

The purpose of this study was to investigate whether *L. plantarum* HAC03 has an anti-obesity effect and whether, when the *L. plantarum* HAC03 was combined with *G. cambogia*, natural substances that possess anti-obesity properties, this enhances their individual anti-obesity effects.

*Lactobacillus plantarum* is a genus of *lactobacillus* spp. that has been extensively studied and used worldwide as a type of probiotics, known to be safe and effective in alleviating various diseases including metabolic diseases such as diabetes and obesity. The safety of *L. plantarum* HAC03 has been verified in previous studies, however, its functional properties have not been investigated.

We found that *L. plantarum* HAC03 has an anti-obesity effect by administering *L. plantarum* HAC03 with a high-fat diet for 10 weeks. The *L. plantarum* HAC03 treatment group (HFD + L1) showed a significantly lower weight gain and weight of four types of adipose tissues (SAT, IAT, MAT, and BAT) than the high-fat diet fed group (HFD) (Figure 1c,d). In addition, the HFD + L1 group reduced weight gain as much as the *L. plantarum* 299V treatment group (HFD + L2), which was used as a positive control. *L. plantarum* 299V has been extensively studied worldwide since it was isolated from the human intestine in 1993 and its anti-obesity effects have been demonstrated [50].

Our objective was to develop a novel anti-obesity combination that would surpass the existing effect by combining *L. plantarum* with a natural compound with a proven anti-obesity effect. *G. cambogia* is a plant known for its anti-obesity effects, with hydroxycitric acid (HCA) being the primary substance that provides anti-obesity effects [51]. In addition, it has an anti-obesity effect through the interaction with gut microbiota [52], although this interaction needs further research.

We found that the combination of *L. plantarum* HAC03 and *G. cambogia* had a greater anti-obesity effect than either substance alone. To determine whether a combination of two substances has a synergistic effect, the concentration of each substance was deliberately set low, and individual or combined treatment with each substance was conducted. After an 11-week animal experiment, our results showed that the group receiving a combination of *L. plantarum* HAC03 and *G. cambogia* extract with a high-fat diet (HFD + L1′ + G) showed a significant reduction in weight gain compared to the HFD group (Figure 2b,c). However, no significant difference was observed between the group receiving low concentration of *L. plantarum* HAC03 only (HFD + L1′) and the group receiving *G. cambogia* only (HFD + G). Based on these results, we found that *L. plantarum* HAC03 requires a minimum intake of 1 × 10^9^ CFU/mouse to have an anti-obesity effect, and that *G. cambogia* requires a specific minimum dosage to produce a significant effect. We found that the HFD + L1′ + G group significantly decreased in three types of adipose tissue (SAT, MAT and BAT). Interestingly, there was no significant reduction in EAT (Figure 2d). Based on the significant weight loss observed in MAT, which is close to the intestine, and the lack of significant weight loss observed in EAT, which is further away from the intestine, we can infer that gut microbiota may play a significant role in alleviating obesity. We found that a combination of *L. plantarum* HAC03 and *G. cambogia* can enhance appetite suppression more than each substance individually. One of the mechanisms by which HCA in *G. cambogia* can contribute to weight management is by increasing serotonin secretion, which suppresses appetite [52]. Although the HFD + G group did not significantly decrease the weekly calorie intake, the HFD + L1′ + G group significantly decreased the weekly calorie intake compared to the HFD group (Figure 2e).

We found that combining *L. plantarum* HAC03 with *G. cambogia* extract has a synergic effect on restoring the biochemical markers of obesity. In addition, this combination has a synergic effect on restoring liver health. The levels of alanine aminotransferase (ALT) and aspartate aminotransferase (AST) decreased more in the HFD + L1′ + G group than the sum of effects observed in the single treatment groups (Table 3). ALT and AST are enzymes that indicate liver function, and their levels increase in the blood when liver cells are damaged. Therefore, measuring the levels of ALT and AST is used as an indicator to assess liver health [53]. There was also a reduction in other biochemical markers of obesity (TG, T-chol, and LDL) in the HFD + L1′ + G group compared to the HFD + L1′ and HFD + G group. The interesting point is that when comparing the single substance treatment groups with the HFD group, we found that the *L. plantarum* HAC03 treatment only group (HFD + L1′) significantly decreased levels of TG, T-chol and LDL. However, the *G. cambogia* treatment only group (HFD + G) did not significantly reduce levels of all biochemical parameters. From the results above, we can infer that although low concentrations of *L. plantarum* HAC03 did not significantly reduce body fat or weight, it had an impact on the biochemical substances in the blood.

We found that the combination of *L. plantarum* HAC03 and *G. cambogia* has a greater decreasing gene expression of fatty acids synthesis than each substance individually. The major mechanism of *G. cambogia*’s anti-obesity effect is the inhibition of ATP-citrate lyase (ACL) by HCA, which binds to ACL instead of citric acid, thus reducing the production of acetyl CoA. If acetyl CoA decreases, the synthesis of malonyl CoA by acetyl-CoA carboxylase (ACC) will decrease, ultimately leading to a reduction in fatty acid synthesis by fatty acid synthase (FAS) [54]. The gene expression levels of the peroxisome proliferator-activated receptor γ (PPARγ) and sterol regulatory element-binding protein 1 (SREBP1c) are correlated with fatty acid synthesis [55,56]. We found that the HFD + L1′ + G group showed significant reductions in the levels of FAS, ACC, PPARγ, and SREBP1c expression, all of which are involved in fatty acid synthesis, compared to the HFD, HFD + L1′ and HFD + G groups in MAT, which reduced the weight and size of adipose tissue (Figure 4).

Globally, a number of studies have revealed that obesity is closely related to the imbalance of gut microbiota [57]. The imbalance of gut microbiota can be both the cause and result of obesity, and probiotics can alleviate obesity by restoring the imbalance [58]. To evaluate the effects of complex *L. plantarum* HAC03 and *G. cambogia* on gut microbiota, a gut microbiome analysis was conducted based on the 16S rRNA gene. We found that gut microbiota composition varies depending on the orally administered substance. Alpha diversity is represented by showing the number of species in a particular ecosystem [59]. A comparison of alpha diversity values between obese and normal people revealed that the alpha diversity values of obese patients were lower than those of normal people [60]. Consistent with previous studies, we observed that the alpha diversity of the HFD group was lower than the LFD group (Figure 5a). In the HFD + L1′ group, the number and evenness of gut microbiota increased compared with the HFD group. On the other hand, in the HFD + G group, the number and evenness of gut microbiota decreased, and the variation between samples increased significantly. *G. cambogia* has been reported to have antimicrobial effects on both gram-positive and gram-negative bacteria [61]. Due to its broad-spectrum antimicrobial properties, *G. cambogia* may have a negative impact on gut microbiota itself, despite its positive effects on weight loss.

Based on the results, we can infer that the combination of *L. plantarum* HAC03 and *G. cambogia* extract can restore the dysbiosis caused by a high-fat diet. In the figure analyzed using PCoA2 and PCoA3, all groups receiving *L. plantarum* HAC03 or *G. cambogia*, whether administered alone or in combination (HFD + L1′, HFD + G, and HFD + L1′ + G), were located between the LFD and HFD groups. In particular, the HFD + L1′ + G group was closest to the LFD group (Figure 5b). Beta diversity quantifies the number of different communities in the region. It provides information about the degree of differentiation among biological communities [62]. When there is a greater difference between groups, the distance between them is greater, while when there is a smaller difference between groups, the distance between them is closer.

We found that specific bacteria related to *L. plantarum* HAC03 and *G. cambogia* extract treatment individually or together. Furthermore, since these bacteria have been demonstrated to alleviate obesity and related metabolic diseases, it is likely that when these two substances are treated individually or in combination, they will have an anti-obesity impact by regulating gut microbiota. Our results showed that the HFD + L1′ + G group, *Eubacterium coprostanoligenes* group and *Lachnospiraceae* UCG 010 group were significantly increased compared to other groups (Figure 5d). *Eubacterium* and *Lachnospiraceae* are the main producers of short chain fatty acids including butyrate and propionate [63,64,65,66]. The *Eubacterium coprostanoligenes* group can alleviate metabolic disease by decreasing the cholesterol concentration in the blood [67]. *Lachnospiraceae* helps to alleviate metabolic disease by using diet-derived polysaccharides, including starch [68]. Based on the results, we have inferred that the *Eubacterium coprostanoligenes* group and *Lachnospiraceae* are specifically correlated to the complex of *L. plantarum* HAC03 and *G. cambogia* extract. We found that the HFD + L1′ group showed a significant increase in *Butyricicoccus* (Appendix A). Butyrate has several beneficial properties that are essential to maintaining gastrointestinal health. It is the main energy source of the colonocytes, inhibits pro-inflammatory pathways, reduces the oxidative stress in the colon and maintains gut barrier function [69,70]. Therefore, butyrate-producing bacteria are seen as the next generation of probiotics. *Butyricicoccus* can produce butyrate. There are several studies showing that consumption of *lactobacillus plantarum* leads to an increase in *Butyricicoccus* [71,72]. We found that the HFD + G group showed a significant increase in *Bacteroides*, *Parabacteroides*, and *Defluviitaleaceae* UCG 011(Appendix A). *Bacteroides* and *Parabacteroides* are negatively correlated with obesity. Among the types of *Parabacteroides, Parabacteroides goldsteinii* can prevent and treat obesity and the related metabolic disorder [73]. In addition, it increases when treated with *garcinia dulcis* [74]. *Defluvitaleaceae* UCG-011 was associated with lipid metabolism (primary bile acid biosynthesis, steroid biosynthesis, fatty acid degradation, and biosynthesis of unsaturated fatty acids) and endocrine metabolic disease (type II diabetes mellitus) [75].

We found functional abundances of each group based on the 16S rRNA gene, and inferred that the combination of *L. plantarum* HAC03 and *G. cambogia* treatment has a superior anti-obesity effect compared to individual treatment. We found that the HFD + L1′ + G group significantly increased the NAD salvage pathway I and starch degradation V functions (Figure 5f). The nicotinamide adenine dinucleotide (NAD+) salvage pathway is a biochemical pathway that allows cells to recycle and maintain the levels of the coenzyme NAD+. NAD+ is a critical coenzyme involved in various metabolic processes in cells, including energy production, DNA repair, and the regulation of gene expression [76]. NAD can boost metabolism, helping provide enough fuel to burn fat reserves to power the rest of the body [77]. As mentioned earlier, the gut microbiota of the *G. cambogia* treatment group predicted an increase in the function of energy production through glycolysis or the citric acid cycle, and the gut microbiota of the *L. plantarum* HAC03 treatment group predicted an increase in BCAA synthesis. We found that the HFD + L1′ group increased in branched chain amino acid (BCAA) synthesis functions, such as L-isoleucine biosynthesis, and pyruvate fermentation to isobutanol (Appendix A). Based on these results, we can infer that gut microbiota of the *L. plantarum* HAC03 treatment group increase in BCAA synthesis. Some *Lactobacillus* strains can transfer to BCAAs [78]. Leucine, isoleucine, and valine belong to the group of amino acids known as BCAAs. BCAAs can aid in weight loss by enhancing fat metabolism and promoting muscle synthesis [79]. The HFD + L1′ group increases pyruvate fermentation to the isobutanol function. Lactic acid bacteria, including lactobacillus, can convert pyruvate to lactate through lactate dehydrogenase [80]. Based on this result, we can infer that if pyruvate is converted to acetate or lactate, the pyruvate used for fatty acid synthesis may decrease, potentially leading to a decrease in fatty acid synthesis. We found that the HFD + G group showed a significant increase in the super pathway of thiamin diphosphate biosynthesis II, thiazole biosynthesis I, and D-galacturonate degradation functions (Appendix A). Based on these results, we can infer that gut microbiota of the *G. cambogia* treatment group increase the function of energy production through glycolysis or the citric acid cycle. Thiamine diphosphate (TDP) plays a crucial role in glycolysis and the citric acid cycle by activating TDP-dependent enzymes, including transketolase (TK), pyruvate dehydrogenase (PDH), and alpha-ketoglutaric acid dehydrogenase (AKGDH), in conjunction with magnesium [81]. Thiazole derivatives have been reported to exhibit anti-diabetic effects through the inhibition of ɑ-glucosidase and the inhibition of protein tyrosine phosphatase 1B (PTP1B) [82,83]. By inhibiting ɑ-glucosidase, thiazole can slow down the absorption of carbohydrates and prevent postprandial hyperglycemia. D-galacturonate is a sugar acid that is derived from the polysaccharide pectin [84]. Pectin is a complex polysaccharide that is found in the cell walls of plants including *G. cambogia* [85]. Through the degradation process, D-galacturonate is broken down into pyruvate and D-glyceraldehyde-3-phosphate, and pyruvate can be converted into acetyl-CoA, which can be used to generate energy in the citric acid cycle. Based on these results, we can infer that gut microbiota in the *G. cambogia* treatment group increase the energy production by activating glycolysis and the citric acid cycle. Ultimately, these predicting functions may help to alleviate obesity by converting stored fat into energy. These results can be inferred from the combination of the two substances. Based on our prediction, we can infer that a combination of two compounds can produce more energy by enhancing the function of the NAD salvage pathway, and more BCAA or SCFA can be produced by enhancing the function of starch degradation. Thus, the enhancement of these functions will act as a mechanism that will exhibit a more potent anti-obesity effect. 

In this study, we discovered that *L. plantarum* HAC03 has an anti-obesity effect, and, when combined with *G. cambogia*, a natural substance already known to provide an anti-obesity effect, provides a new anti-obesity effect beyond what either substance could provide alone. The combination of *L. plantarum* HAC03 and *G. cambogia* extract significantly reduced adipocyte size, adipose tissue weight, and body weight with decreasing gene expression of fatty acid synthesis compared to the two substances alone. Furthermore, this enhanced anti-obesity effect could be achieved by altering the composition of specific gut microbiota, such as the *Eubacterium coprostanoligenes* group and *Lachnospiraceae* UCG 010 group, which would enhance energy production and increase BCAA and SCFA levels. This combination represents a new anti-obesity candidate material that surpasses the limitations of probiotics on their own.

## Figures and Tables

**Figure 1 nutrients-15-01859-f001:**
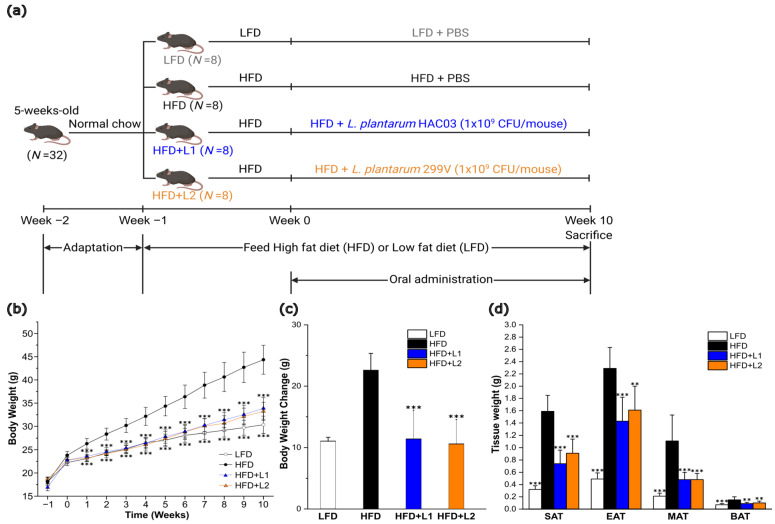
*L. plantarum* HAC03 has an anti-obesity effect. (**a**) Scheme of animal experiment. LFD: Low-fat diet, HFD: High-fat diet, HFD + L1: High-fat diet with *L. plantarum* HAC03 treatment, HFD + L2: High-fat diet with *L. plantarum* 299V treatment. (**b**) Body weight changes during the animal experiment. (**c**) Weight gain during the animal experiment. (**d**) Adipose tissue weight after 11 weeks of *L. plantarum* HAC03 or *L. plantarum* 299V treatments. The data are presented as mean ± SD (*n* = 8). A one-way ANOVA with Dunnett’s test was used for comparison with HFD groups. ** *p* < 0.01, *** *p* < 0.001.

**Figure 2 nutrients-15-01859-f002:**
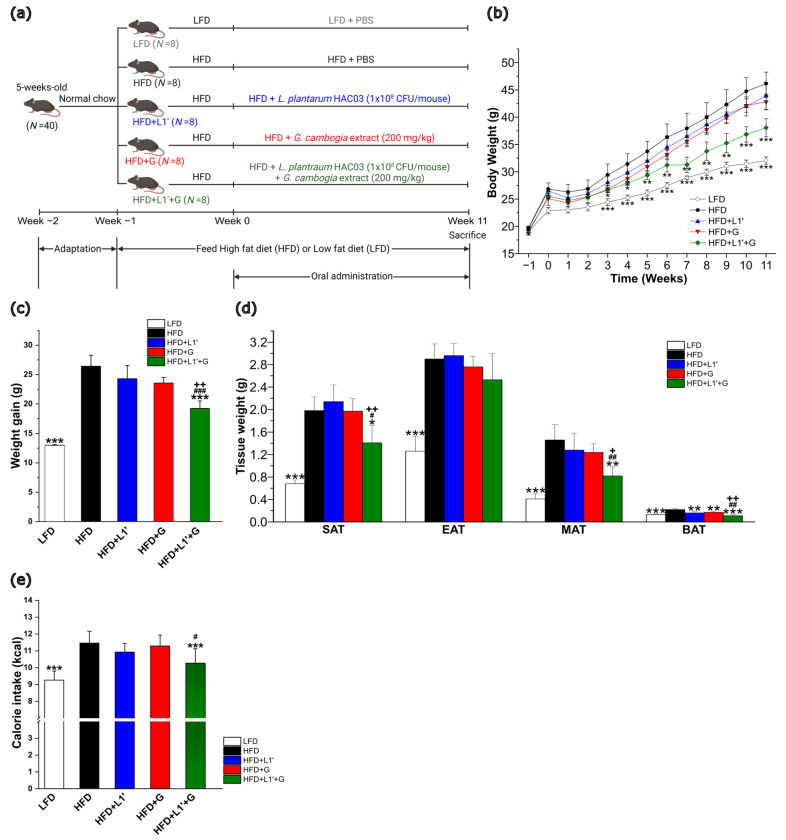
Combining *L. plantarum* HAC03 and *G. cambogia* has anti-obesity synergic effects. (**a**) Scheme of animal experiment. LFD: Low-fat diet, HFD: High-fat diet, HFD + L1′: High-fat diet with low-dose *L. plantarum* HAC03 treatment, HFD + G: High-fat diet with *G. cambogia* treatment, HFD + L1′ + G: High-fat diet with low-dose *L. plantarum* HAC03 and *G. cambogia* treatment. (**b**) Body weight changes during the animal experiment. (**c**) Weight gain during the animal experiment. (**d**) Tissue weight after 11 weeks of treatments. (**e**) Weekly calorie intake. The data are presented as mean ± SD (*n* = 8). A one-way ANOVA with Dunnett’s comparison test was used for comparison with HFD groups. * *p* < 0.05, ** *p* < 0.01, *** *p* < 0.001. A Student’s two-tailed *t*-test was used for analysis of difference between experimental groups. # *p* < 0.05, ## *p* < 0.01, ### *p* < 0.001 between HFD + G group and HFD + L1′ + G group. + *p* < 0.05, ++ *p* < 0.01 between HFD + L1′ group and HFD + L1′ + G group.

**Figure 3 nutrients-15-01859-f003:**
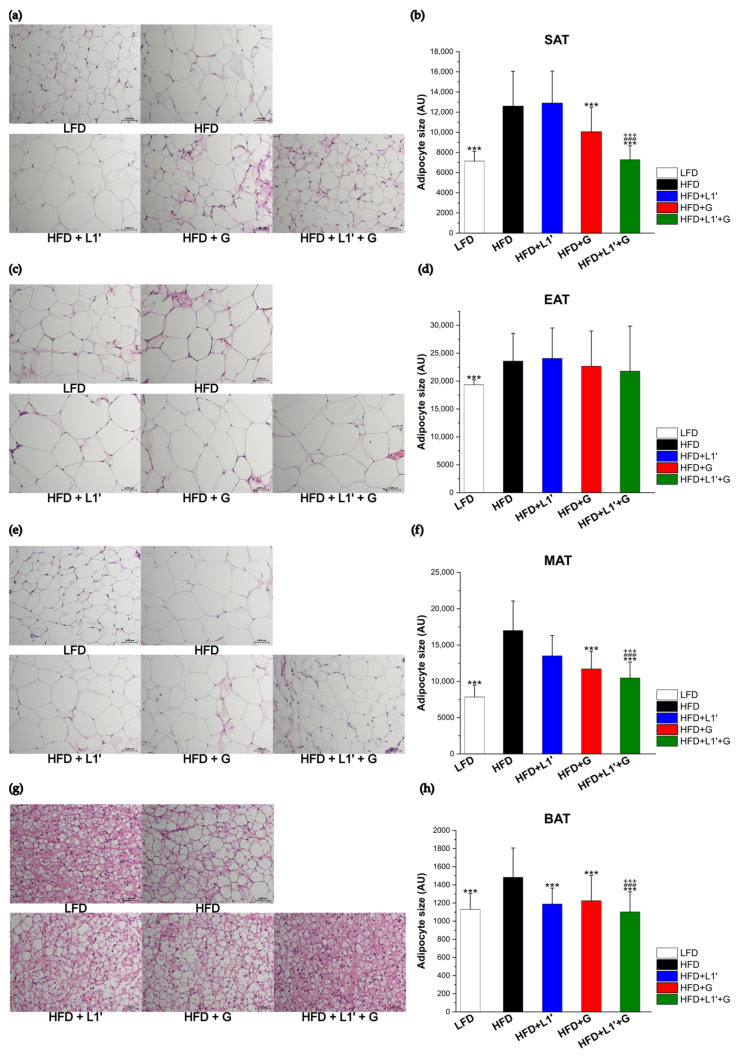
Combining *L. plantarum* HAC03 and *G. cambogia* has a synergic effect on decreasing the adipocyte size. (**a**,**c**,**e**,**g**) Histological features of adipose tissue between groups. Representative photomicrographs of adipose tissue sections stained with hematoxylin and eosin (×200) are shown. Subcutaneous adipose tissue (SAT), epididymal adipose tissue (EAT), mesenteric adipose tissue (MAT), and brown adipose tissue (BAT). (**b**,**d**,**f**,**h**) Adipocyte size of adipose tissue between groups. The data are presented as mean ± SD (*n* = 8). A one-way ANOVA with Dunnett’s comparison test was used for comparison with HFD groups. *** *p* < 0.001. A Student’s two-tailed *t*-test was used for analysis of difference between experimental groups. ### *p* < 0.001 between HFD + G group and HFD + L1′ + G group. +++ *p* < 0.001 between HFD + L1′ group and HFD + L1′ + G group.

**Figure 4 nutrients-15-01859-f004:**
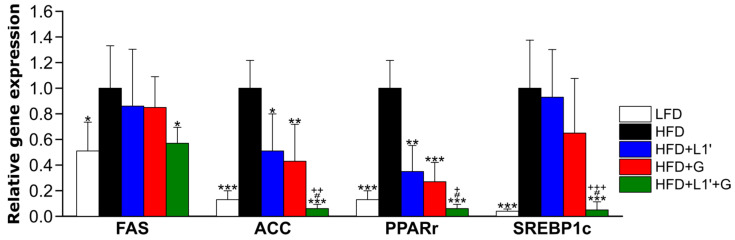
Combining *L. plantarum* HAC03 and G. cambogia has a synergic effect on decreasing mRNA expression of fatty acid synthesis. Relative gene expression related to fatty acid synthesis in the MAT. Fatty acid synthase, FAS; acetyl-CoA carboxylase, ACC; peroxisome proliferator activated receptor gamma, PPARγ; sterol regulatory element binding protein 1c, SREBP1c. All genes are normalized to expression of β-actin. The data are presented as mean ± SD (*n* = 8). A one-way ANOVA with Dunnett’s comparison test was used for comparison with HFD groups. * *p* < 0.05, ** *p* < 0.01, *** *p* < 0.001. A Student’s two-tailed *t*-test was used for analysis of difference between experimental groups. # < 0.05 between HFD + G group and HFD + L1′ + G group. + < 0.05, ++ < 0.01, +++ < 0.001 between HFD + L1′ group and HFD + L1′ + G group.

**Figure 5 nutrients-15-01859-f005:**
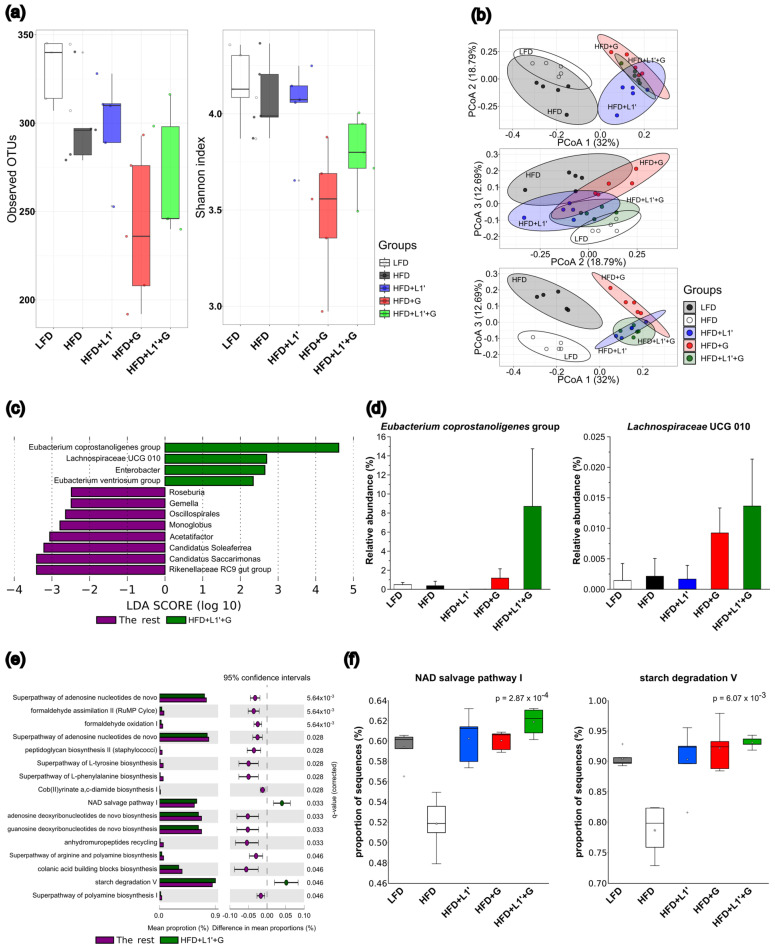
The combination of *L. plantarum* HAC03 and *G. cambogia* affect gut microbiota modulation. (**a**) alpha diversity. (**b**) beta diversity. (**c**) Linear discriminant analysis (LDA) effect size (LEfSe) with LDA effect size ≥ 2 and alpha ≤ 0.05 for HFD + L1′ + G group. (**d**) gut microbiota related HFD + L1′ + G group. (**e**) Functional potential prediction of HFD + L1′ + G group and the rest. (**f**) Specific functions correlated HFD + L1′ + G group. The data presented as mean ± SD (*n* = 5). Significance for data is calculated using a one-way ANOVA followed by a Bonferroni multiple comparison test.

**Table 1 nutrients-15-01859-t001:** All groups of animal experiments.

Groups	Feed Type	Treatment
LFD	Low fat diet	200 µL PBS
HFD	High fat diet	200 µL PBS
HFD + L1	High fat diet	*L. plantarum* HAC03 (1 × 10^9^ CFU/mouse)
HFD + L2	High fat diet	*L. plantarum* 299V (1 × 10^9^ CFU/mouse)
HFD + L1′	High fat diet	*L. plantarum* HAC03 (1 × 10^8^ CFU/mouse)
HFD + G	High fat diet	*G. cambogia* (200 mg/kg)
HFD + L1′ + G	High fat diet	*L. plantarum* HAC03 (1 × 10^8^ CFU/mouse) + *G. cambogia* (200 mg/kg)

**Table 2 nutrients-15-01859-t002:** Primer sequences for qPCR.

Gene	Forward Primer 5′-3′	Reverse Primer 5′-3′
FAS	CTGGACTCGCTCATGGGTG	CATTTCCTGAAGTTTCCGCAG
ACC	TGACAGACTGATCGCAGAGAAAG	TGGAGAGCCCCACACACA
PPARγ	AGTGGAGACCGCCCAGG	GCAGCAGGTTGTCTTGGATGT
SREBP1c	AGCAGCCCCTAGAACAAACAC	CAGCAGTGAGTCTGCCTTGAT

**Table 3 nutrients-15-01859-t003:** Combining *L. plantarum* HAC03 and *G. cambogia* has a synergic effect on decreasing the biochemical markers of obesity.

Parameter	Groups				
	LFD	HFD	HFD + L1′	HFD + G	HFD + L1′ + G
ALT (U/L)	34.5 ± 6.59 ***	70.8 ± 9.80	69.1 ± 12.84	60.0 ± 9.00	36.1 ± 4.84 *** +++/###
AST (U/L)	78.7 ± 6.83 ***	110.2 ± 10.06	108.5 ± 7.04	88.7 ± 7.79 ***	72,6 ± 11.22 *** +++/#
TG (mg/dL)	65.0 ± 4.08 ***	122.5 ± 18.87	95.0 ± 22.91 *	100.0 ±23.73	78.3 ± 8.50 ***
T-chol (mg/dL)	118.3 ± 8.22 ***	171.0 ± 17.68	147.3 ± 8.65 *	160.7 ± 4.92	138.3 ± 3.09 ** ###
HDL (mg/dL)	60.4 ± 4.64 ***	73.0 ± 1.13	69.5 ± 5.95	74.7 ± 0.85	74.2 ± 1.00
LDL (mg/dL)	9.4 ± 0.98 **	13.9 ± 2.24	9.8 ± 2.14 *	11.7 ± 0.22	9.3 ± 1.13 ** ###

Note: ALT: Alanine aminotransferase, AST: Aspartate aminotransferase, TG: Triglyceride, T-chol: Total cholesterol, HDL: High Density Lipoprotein Cholesterol, LDL: Low Density Lipoprotein Cholesterol. The data are presented as mean ± SD (*n* = 8). A one-way ANOVA with Dunnett’s comparison test was used for comparison with HFD groups. * *p* < 0.05, ** *p* < 0.01, *** *p* < 0.001. A Student’s two-tailed *t*-test was used for analysis of difference between experimental groups. # *p* < 0.05, ### *p* < 0.001 between HFD + G and HFD + L1′ + G. +++ *p* < 0.001 between HFD + L1′ and HFD + L1′ + G.

## Data Availability

Not applicable.

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
