# Peer review of "Combination of Lactobacillus plantarum HAC03 and Garcinia cambogia Has a Significant Anti-Obesity Effect in Diet-Induced Obesity Mice"

_nutrients, 2023, doi:10.3390/nu15081859_

Round 1
Reviewer 1 Report
On line 85, would it be interesting to describe what white kimchi is?
On line 93 write correctly the taxon Garcinia cambogia Roxb.
On line 104 the authors use a probiotic strain or combination of probiotics?
From line 110 to 113 there is a paragraph that should be better placed in the conclusion than in the introduction. It is the conclusion drawn from the results of the research.
The study uses a dosage 1x109/mouse of the probiotic strain that is recommended for humans. Were there no adverse effects?
The source of the Garcinia cambogia Roxb. extract is not specified in the materials and methods section. Neither the concentration of HCA (in point 2.2 it does). Neither the drug/extract ratio. Neither mg of extract. Could the authors describe it?.
Have any studies been conducted to see how G. cambogia Roxb. extract might affect the microbiota due to its antimicrobial properties?
Conclusion
This study is very interesting. A formulation of this probiotic/extract association that is stable over time should be developed. In this way, it could be a therapeutic tool with fewer adverse effects than liraglutide, semaglutide, etc.
Author Response
Point 1. On line 85, would it be interesting to describe what white kimchi is?
Response 1: White kimchi is one of the traditional fermented foods of South Korea.
Line 86 has been changed
Lactobacillus plantarum HAC03 used in this study is a strain identified from the white kimchi, which is one of the traditional fermented foods of South Korea.
Point 2. On line 94 write correctly the taxon Garcinia cambogia Roxb.
Response 2: We change from garcinia cambogia to Garcinia cambogia
Point 3. On line 104 the authors use a probiotic strain or combination of probiotics?
Response 3: We use a probiotic strain not the combination of probiotics.
Line 105 has been changed
We hypothesized that a combination of probiotic strain which has anti-obesity effects and G. cambogia with proven anti-obesity, will exhibit a synergistic effect between the two substances, resulting in a greater anti-obesity effect than when each substance is used alone.
Point 4. From line 110 to 113 there is a paragraph that should be better placed in the conclusion than in the introduction. It is the conclusion drawn from the results of the research.
Response 4: Lines 110 to 113 have been deleted.
Point 5. The study uses a dosage 1x109/mouse of the probiotic strain that is recommended for humans. Were there no adverse effects?
Response 5: There were no significant adverse effects observed during the experimental period.
The lines 205 to 206 have been added to the paragraph (Results 3.1.)
During the 10 week animal experiment, there were no significant adverse effects observed.
Point 6. The source of the Garcinia cambogia Roxb. extract is not specified in the materials and methods section. Neither the concentration of HCA (in point 2.2 it does). Neither the drug/extract ratio. Neither mg of extract. Could the authors describe it?.
Response 6: The sentence of Materials and Methods section 2.1. (Line 116) has been changed.
2.1. Bacterial strains, culture conditions and Natural product.
Information of G. cambogia has been added in paragraph of materials and methods section 2.1. (line 123 to 125).
- cambogia extracts (main component: hydroxycitric acid, 65%) was obtained from DAEUNE FS CO., Ltd. (Ansan-si, Gyeonggi-do, South Korea).
Point 7. Have any studies been conducted to see how G. cambogia Roxb. extract might affect the microbiota due to its antimicrobial properties?
Response 7. It has been confirmed through various studies that Garcinia cambogia possesses antimicrobial properties against both gram-positive and gram-negative bacteria, as mentioned in the discussion section (line 470 to 471).

Reviewer 2 Report
In this study, the author utilized a diet-induced obesity mouse (DIO) model to investigate the effects of combining Lactobacillus plantarum HAC03 with Garcinia cambogia extract. The study was well done, but i think the detail information about the G. cambogia extracts should be provided. For example, the prepared method and the chemical components should be provided in the manuscript.
Author Response
Point 1. The reviewer thinks the detail information about the G. cambogia extracts should be provided. For example, the prepared method and the chemical components should be provided in the manuscript.
Response 1: The sentence of Materials and Methods section 2.1. (Line 116) has been changed.
2.1. Bacterial strains, culture conditions and Natural product
Information of G. cambogia has been added in paragraph of materials and methods section 2.1. (line 123 to 125).
G. cambogia extracts (main component: hydroxycitric acid, 65%) was obtained from DAEUNE FS CO., Ltd. (Ansan-si, Gyeonggi-do, South Korea).
